# Slack Potassium Channels Modulate TRPA1-Mediated Nociception in Sensory Neurons

**DOI:** 10.3390/cells11101693

**Published:** 2022-05-19

**Authors:** Fangyuan Zhou, Katharina Metzner, Patrick Engel, Annika Balzulat, Marco Sisignano, Peter Ruth, Robert Lukowski, Achim Schmidtko, Ruirui Lu

**Affiliations:** 1Institute of Pharmacology and Clinical Pharmacy, Goethe University, 60438 Frankfurt am Main, Germany; zhou@stud.uni-frankfurt.de (F.Z.); metzner@em.uni-frankfurt.de (K.M.); p.engel@em.uni-frankfurt.de (P.E.); balzulat@em.uni-frankfurt.de (A.B.); schmidtko@em.uni-frankfurt.de (A.S.); 2Institute of Clinical Pharmacology, Pharmazentrum Frankfurt/ZAFES, Goethe University, 60590 Frankfurt am Main, Germany; marco.sisignano@med.uni-frankfurt.de; 3Department of Pharmacology, Toxicology and Clinical Pharmacy, University of Tübingen, 72076 Tübingen, Germany; peter.ruth@uni-tuebingen.de (P.R.); robert.lukowski@uni-tuebingen.de (R.L.)

**Keywords:** TRPA1, slack, dorsal root ganglia, pain, mice

## Abstract

The transient receptor potential (TRP) ankyrin type 1 (TRPA1) channel is highly expressed in a subset of sensory neurons where it acts as an essential detector of painful stimuli. However, the mechanisms that control the activity of sensory neurons upon TRPA1 activation remain poorly understood. Here, using in situ hybridization and immunostaining, we found TRPA1 to be extensively co-localized with the potassium channel Slack (K_Na_1.1, Slo2.2, or Kcnt1) in sensory neurons. Mice lacking Slack globally (Slack^−/−^) or conditionally in sensory neurons (SNS-Slack^−/−^) demonstrated increased pain behavior after intraplantar injection of the TRPA1 activator allyl isothiocyanate. By contrast, pain behavior induced by the TRP vanilloid 1 (TRPV1) activator capsaicin was normal in Slack-deficient mice. Patch-clamp recordings in sensory neurons and in a HEK cell line transfected with TRPA1 and Slack revealed that Slack-dependent potassium currents (I_KS_) are modulated in a TRPA1-dependent manner. Taken together, our findings highlight Slack as a modulator of TRPA1-mediated, but not TRPV1-mediated, activation of sensory neurons.

## 1. Introduction

Noxious stimuli are detected by sensory neurons of the dorsal root ganglia (DRG) or trigeminal ganglia that convey the nociceptive information from peripheral sites to the dorsal horn of the spinal cord or spinal trigeminal nucleus. There the inputs are synaptically transmitted onto interneurons, which further integrate and relay the incoming sensory information to various brain regions [1,2]. While the ability to detect noxious stimuli is essential for wellbeing and survival, alterations of the pain pathway may lead to hypersensitivity, such that pain outlives its usefulness as an acute warning system and may become chronic [3]. In fact, chronic pain is a big health issue with a complex etiology and heterogeneity that represents a pressing challenge to neuroscientists and clinicians. Thus, understanding the basic mechanisms of pain processing will help to improve the treatment of chronic pain in future [4,5,6].

Various detectors and transducers of noxious stimuli have been identified in the past decades. These include specialized receptors belonging to the transient receptor potential (TRP) family of ligand-gated ion channels, which are predominantly expressed by distinct subsets of sensory neurons [7,8,9,10,11]. The calcium-permeable nonselective cation channel TRP ankyrin 1 (TRPA1) is a low-threshold sensor for a variety of chemical compounds, which includes numerous natural pungent smells, such as allyl isothiocyanate (AITC), an ingredient in mustard oil and wasabi, or allicin, one of the main active compounds in garlic [12,13,14,15,16]. Other TRPA1 activators include formaldehyde, acrolein, menthol, cannabidiol, nicotine, and various endogenous compounds that are released during oxidative stress and inflammation [9,17,18,19]. Once TRPA1 is activated, it contributes to membrane depolarization, action potential generation, and neurotransmitter release [20,21]. Numerous studies demonstrated that TRPA1 plays a key role as a sensor of chemical-, heat-, and cold-induced pain, is involved in the processing of acute itch, inflammatory, and neuropathic pain [7,8,19,22,23,24,25,26,27], and has long been proposed that blockade of TRPA1 could be a therapeutic strategy to treat pain and other diseases [7,23,26,28]. However, even though various interactors of TRPA1 in sensory neurons have been identified [29,30,31], it remains poorly understood how the activity of TRPA1-positive (TRPA1^+^) neurons is controlled after activation.

Recent single-cell RNA-sequencing (scRNA-seq) studies suggest that in sensory neurons, there is a substantial expressional overlap of TRPA1 and the Na^+^-activated K^+^ channel Slack (also termed K_Na_1.1; gene *Kcnt1*) [32,33]. In fact, previous immunostaining and in situ hybridization experiments demonstrated that Slack, such as TRPA1 [22,34,35,36], is abundantly expressed in nonpeptidergic C fibers [37,38]. Furthermore, Slack plays a major role in shaping neuronal electrical properties [37,38,39] and regulating neurotransmitter release from sensory neurons [40]. These properties led us to speculate that Slack might modulate the activity of TRPA1^+^ sensory neurons. Therefore, we here investigated the extent of co-localization of TRPA1 and Slack and the potential functional interaction of these two channels in sensory neurons.

## 2. Materials and Methods

### 2.1. Animals

The generation of global Slack-deficient mice (Slack^−/−^) and of tissue-specific Slack mutants lacking Slack in Na_V_1.8-positive sensory neurons (SNS-Slack^−/−^; littermate floxed Slack mice were used as a control) on a C57BL/6 background was described earlier [37]. To ablate Slack selectively in dorsal horn neurons, floxed Slack mice were crossed with Lbx1-Cre mice [41] to obtain homozygous conditional Slack knockouts (Lbx1-Slack^−/−^; littermate floxed Slack mice were used as a control). Mice had free access to food and water and were housed on a 12/12 light/dark cycle. Experiments were performed in 6- to 12-week-old animals using approximately equal numbers of male and female mice [42,43]. We analyzed sex as a variable in all mouse experiments, but we did not detect significant main effects of sex in any test in this study. In total, 178 mice were used in this study. All experiments were performed according to the International Association for the Study of Pain (IASP) guidelines, the ARRIVE (Animal research: Reporting of In Vivo Experiments) guidelines, and the 3Rs principles and were approved by the local Ethics Committee for Animal Research (Darmstadt, Germany; numbers V54-19c20/15-FR/1013 and V54-19c18-FR/2005).

### 2.2. Behavioral Testing

All behavioral testing was performed between 8 am and 6 pm in a room maintained at 22 ± 2 °C and 55 ± 10% humidity. Before the test day, mice were acclimated to the experimental room for at least two consecutive days. Prior to the experiments, mice were habituated to the experimental apparatus for at least 30 min. All behavioral experiments were performed by an investigator who was blinded to the genotype of the mice.

#### 2.2.1. Mechanical Sensitivity

Mechanical sensitivity thresholds were measured as described previously [39]. Briefly, mice were individually confined in boxes placed on an elevated metal mesh floor. Calibrated von Frey filaments ranging from 0.02 to 1.4 g (Ugo Basile, Gemonio, Italy) were applied to the plantar surface of a hind paw. Clear paw withdrawal, shaking, or licking during or immediately following the stimulus (up to 3 s after the filament was bowed) was considered a nociceptive response. The 0.6 g filament was the first stimulus to be used. The 50% withdrawal thresholds were determined using the up-down method and calculated using the online tool “Up-down method for von Frey experiments” (https://bioapps.shinyapps.io/von_frey_app/ (accessed on 12 October 2021)) [44,45]. Per animal, 3 measurements were performed (>5 min between each measurement) and averaged.

#### 2.2.2. AITC-Induced Pain Behavior

The TRPA1 activator allyl isothiocyanate (AITC; 10 mM in 20 µL saline containing 0.05 or 2% dimethyl sulfoxide (DMSO), Sigma-Aldrich, Darmstadt, Germany) was intraplantarly injected into the hind paw [46], and the time spent licking and biting the injected paw was recorded in a Plexiglass cylinder for 30 min using a full HD camera (HC-V380; Panasonic, Kadoma Osaka, Japan). Immediately thereafter, mice were placed in boxes on an elevated metal mesh floor, and mechanical sensitivity was evaluated over 72 h.

#### 2.2.3. Capsaicin-Induced Pain Behavior

Capsaicin (5 µg in 20 µL saline containing 2% DMSO, Sigma Aldrich) was injected into a hind paw [33], and the time spent licking and biting the injected paw was recorded in a Plexiglass cylinder for 10 min using a full HD camera. Immediately thereafter, mice were placed in boxes on an elevated metal mesh floor, and mechanical sensitivity was evaluated over 72 h.

### 2.3. qRT-PCR

Mice were euthanized by carbon dioxide (CO_2_) inhalation and lumbar (L3–L5) DRGs, lumbar (L3–L5) spinal cord, cerebellum and prefrontal cortex were rapidly dissected, snap frozen in liquid nitrogen and stored at −80 °C until use. Total RNA from the spinal cord, cerebellum, and prefrontal cortex was extracted using TRIzol reagent (#15596026; Thermo Fisher Scientific, Frankfurt, Germany) or QIAzol lysis reagent (#79306; Qiagen, Hilden, Germany) and chloroform in combination with the RNeasy Mini Kit (#74104; Qiagen) according to the manufacturer’s recommendations. Total RNA from DRGs was isolated using the innuPREP Micro RNA Kit (#C-6134; Analytik Jena, Berlin, Germany) following the manufacturer’s instructions.

Isolated RNA was quantified with a NanoDrop 2000 (Thermo Fisher Scientific), and cDNA was synthesized from 200 ng using the first strand cDNA synthesis kit (#10774691; Thermo Fisher Scientific) with random hexamer primer. Quantitative real-time reverse transcription PCR (qRT-PCR) was performed on a CFX96 Touch Real-Time System (Bio-Rad, Hercules, Germany) using the iTaq Universal SYBR Green SuperMix (#1725120; Bio-Rad) and primer pairs (Biomers, Ulm, Germany) for Slack (fwd 5′-ctgctgtgcctggtcttca-3′, rev 5′-aaggaggtcagcaggttcaa-3′), TRPA1 (fwd 5′-ggaaataccccactgcattgt-3′, rev 5′-cagctatgtgaaggggtgaca-3′), and glyceraldehyde 3-phosphate dehydrogenase (GAPDH; fwd 5′-caatgtgtccgtcgtggatct-3′, rev 5′-gtcctcagtgtagcccaagatg-3′). qRT-PCR of TRPV1 mRNA was performed using Taqman gene expression assays (Applied Biosystems, San Mateo, CA, USA) for TRPV1 (catalog #Mm01246302_mL) and GAPDH (catalog #Mm99999915_gL). Reactions were performed in duplicate or triplicate by incubating for 2 min at 50 °C and 10 min at 95 °C, followed by 40 cycles of 15 s at 95 °C and 60 s at 60 °C. Water controls were included to ensure specificity. Relative expression of target gene levels was determined using the comparative 2^−ΔΔCt^ method and normalized to GAPDH.

### 2.4. In Situ Hybridization and Immunostaining

Mice were euthanized by CO_2_ inhalation and immediately perfused through the ascending aorta with 0.9% NaCl, followed by 1% or 4% paraformaldehyde (PFA) in phosphate-buffered solution (PBS), pH = 7.4. Intact lumbar (L4-5) DRGs and spinal cord columns were dissected and post-fixed 15 min in 1% or 4% PFA in PBS at room temperature, cryoprotected in 20% sucrose in PBS overnight. Tissues were embedded in tissue freezing medium (Tissue-Tek O.C.T. Compound, #4583, Sakura, Torrance, CA, USA) on dry ice, cryostat sectioned at a thickness of 14 µm, collected on Superfrost Plus Adhesion microscope slides (#J1800AMNZ, Epredia, Braunschweig, Germany), allowed to dry at room temperature for 2 h, and stored at −80 °C until use.

To perform in situ hybridization, we used the ViewRNA ISH tissue core kit (#19931 Thermo Fisher Scientific) and the ViewRNA tissue assay blue module (#19932, Thermo Fisher Scientific) [46,47]. Experiments were performed with frozen tissue sections as indicated by the manufacturer. A type 1 probe set designed by Thermo Fisher Scientific for the coding region of mouse Slack (Kcnt1; #VB1-21049) and a type 6 probe set for mouse TRPA1 (#VB6-18246) were used. Briefly, tissue sections were fixed for 16–18 h in 4% PFA in PBS at 4 °C, dehydrated in graded ethanol, washed in PBS and treated with protease QF for 25 min at 40 °C (Thermobrite; Leica, Germany). Then, the Slack type 1 probe and TRPA1 type 6 labeled probe were simultaneously incubated overnight at 40 °C. After consecutive incubation with preamplifier mix QT, amplifier mix QT, alkaline phosphatase (AP)-conjugated probe and AP enhance the solution, the signal was developed via reaction with a fast red substrate (for Slack type 1 probe) and fast blue substrate (for TRPA1 type 6 probe) in the dark. Finally, slides were washed in PBS and coverslipped with Fluoromount G (Southern Biotech, Birmingham, AL, USA).

In immunostaining experiments, slides were washed in PBS, permeabilized for 5 min in PBST (0.1% Triton X-100 in PBS), blocked in 10% normal goat serum (#10000C, Thermo Fisher Scientific) and 3% bovine serum albumin (BSA, #A6003, Sigma-Aldrich, Darmstadt, Germany) in PBS for 1 h at room temperature, and then incubated with primary antibodies diluted in 3% BSA in PBS overnight at 4 °C or for 2 h at room temperature. Primary antibodies used include mouse anti-Slack (1:500; #75-051, NeuroMab, Davis, CA, USA), rabbit anti-TRPV1 (1:800; #ACC-030, Alomone, Jerusalem, Israel), rabbit anti-calcitonin gene related peptide (CGRP; 1:800; #PC205L, Sigma-Aldrich), mouse anti-neurofilament 200 (NF200; 1:2000; #N4142, Sigma-Aldrich), rabbit anti-tyrosine hydroxylase (TH; 1:400; #AB152, Millipore, Temecula, CA, USA), rabbit anti-glutamate decarboxylase 67 (GAD67; 1:500; AB9706, Chemicon, Temecula, CA, USA), and guinea pig anti-protein kinase C isoform γ (PKCγ; 1:800; AB_2571826, Frontier Institute, Tokyo, Japan). In double-labeling experiments, primary antibodies were consecutively incubated. Secondary antibodies (Alexa Fluor 555 conjugated goat anti-mouse IgG1 antibody (#A-21127, Thermo Fisher Scientific), Alexa Fluor 488 conjugated goat anti-rabbit IgG(H+L) antibody (#A-11008, Thermo Fisher Scientific), Alexa Fluor 555 conjugated goat anti-rabbit IgG(H+L) antibody (#A-21429, Thermo Fisher Scientific) and Alexa Fluor 488 conjugated goat anti-guinea pig IgG(H+L) antibody (#A-11073, Thermo Fisher Scientific)) were incubated in PBS at 1:1000 for 2 h at room temperature. Alexa Fluor 488-conjugated *Griffonia simplicifolia* isolectin B4 (IB4; 10 μg/mL, #121411, Thermo Fisher Scientific) in buffer containing 1 mM CaCl_2_⋅2H_2_O, 1 mM MgCl_2_, 1 mM MnCl_2_ and 0.2% Trion X-100, pH = 7.4, was incubated for 2 h at room temperature. After immunostaining, slices were treated with 0.06% Sudan black B (#199664, Sigma-Aldrich) in 70% ethanol for 5 min to reduce autofluorescence [48], then washed in PBS and mounted with Fluoromount G.

Images were acquired at 20× magnification using an Eclipse Ni-U (Nikon, Düsseldorf, Germany) microscope equipped with a monochrome charge-coupled device camera. The raw image files were brightened, contrasted, pseudocolored and superimposed using Adobe Photoshop 2020 software (Adobe Systems, San Jose, CA, USA). Controls for immunostaining were performed by omitting the first and/or the second primary antibodies and by incubating tissues of Slack^−/−^ mice. Controls for in situ hybridization were performed by incubating type 1 and type 6 scramble probes.

For the quantification of Slack mRNA-positive DRG neuron populations, we cut serial sections of lumbar DRGs (L4–L5) from 3 mice. Per animal ≥2 sections at least 100 µm apart with at least 100 cells were counted manually by an observer (346 cells in total). Only cells containing DAPI-positive nuclei and showing clear staining signals above the background level, with a threshold set based on scramble control hybridization, were included.

For quantification of markers of DRG neuron subpopulations, serial sections of lumbar DRGs (L4–L5) from SNS-Slack^−/−^ and WT mice (3–4 mice per genotype) were cut. Per animal ≥2 sections at least 100 µm apart with at least 100 cells were counted manually by an observer (17,455 cells in total). Only cells showing clear staining signals above background level were included. For calculation of the percentage of marker-positive DRG neurons, the total number of DRG neuron somata was counted based on their autofluorescence visualized in the FITC channel.

### 2.5. Ca^2+^ Imaging

A DRG neuron primary cell culture for Ca^2+^ imaging was prepared as described previously [39,49]. Briefly, after mice were euthanized by exposure to gradually increasing concentration of CO_2_ inhalation, lumbar (L1–L6) DRGs were excised and transferred to ice-cold Hank’s balanced salt solution (#14170-088; Gibco/Thermo Fisher Scientific) and then treated with 500 U/mL collagenase IV (#17104019; Thermo Fisher Scientific) and 2.5 U/mL dispase II (#04942078001; Sigma Aldrich) for 90 min and 0.05% Trypsin/EDTA (#25200-056; Gibco/Thermo Fisher Scientific) for 10 min. DRGs were then washed twice with supplemented Neurobasal A Medium (#10888-022; Gibco/Thermo Fisher Scientific) containing L-glutamine (2 mM; #25030-024; Gibco/Thermo Fisher Scientific), gentamicin (50 µg/mL; #10131-027; Gibco/Thermo Fisher Scientific), penicillin streptomycin (100 U/mL; #15140-122; Gibco/Thermo Fisher Scientific), and 10% fetal bovine serum (#F9665; Sigma Aldrich). Then, DRGs were mechanically dissociated by pipetting up and down with a plastic pipette, and isolated cells were seeded onto poly-D-lysine-coated (200 μg/mL, #A-003-E; Millipore) coverslips and cultured overnight at 37 °C in 5% CO_2_ in supplemented Neurobasal A Medium with additional 2% B27 (#17504-044; Gibco/Thermo Fisher Scientific).

Ca^2+^ imaging experiments were performed 20–26 h after DRG preparation. Neurons were loaded with 5 µM Fura-2-AM-ester (#50033; Biotium, Fremont, CA, USA) in Neurobasal A Medium for 45 min at 37 °C. After loading, coverslips were transferred to the perfusion chamber and continuously superfused with a physiological Ringer solution (145 mM NaCl, 1.25 mM CaCl_2_, 1 mM MgCl_2_, 5 mM KCl, 10 mM glucose and 10 mM HEPES; pH 7.4, adjusted with NaOH) at a flow rate of 1–2 mL/min. Fluorescence was measured during alternating illumination at 340 and 380 nm using a Nikon Eclipse Ts2R inverse microscope equipped with an illumination system (DG4, Sutter Instruments, Novato, CA, USA), a digital camera (ORCA-05G, Hamamatsu, Shizuoka Prefecture, Japan), Fura-2 filters, and a motorized microscope stage (Märzhäuser Wetzlar, Wetzlar, Germany). For a functionality test of TRPA1, the TRPA1 agonist AITC (200 μM) dissolved in Ringer solution was applied by bath perfusion for 15 s. At the end of each measurement, viable neurons were identified by stimulating with 75 mM KCl for 20 s. A positive Ca^2+^ response was defined from a simultaneous increase at 340 nm and a decrease at 380 nm when the fluorescence ratio of 340 nm divided by 380 nm (F340/F380) normalized to baseline exceeded 20% of the baseline level. All experiments were performed at room temperature. Acquired images were displayed as the ratio of F340/F380.

### 2.6. Patch-Clamp Recordings

Lumbar (L1–L5) DRGs from wildtype (WT) and Slack^−/−^ mice were removed, and primary DRG neuron cultures were prepared as previously reported [37]. Human embryonic kidney 293 (HEK293) cells and a HEK293 stable cell line stably expressing human Kcnt1 (referred to as HEK-Slack cells; SB-HEK-KCa4.1; SB Drug Discovery, Lanarkshire, UK) were cultured in minimum essential medium with 10% fetal calf serum, supplemented with 2 mM L-glutamine and 0.6 mg/mL G-418 (all from Gibco/Thermo Fisher Scientific) at 37 °C in humidified 5% CO_2_. HEK-Slack cells that additionally express TRPA1 (referred to as HEK-Slack-TRPA1 cells) were carried out with a pcDNA3.1^+^ vector plasmid (NM_007332.2, GenScript, Leiden, the Netherlands) combined with a GFP-tagged reporter plasmid (pIRES2-EGFP) using Roti-Fect transfection reagent (Carl Roth, Karlsruhe, Germany) according to the manufacturer’s instructions. Cells were seeded on poly-D-Lysine coated coverslips 1 day before experiments.

For whole-cell voltage clamp recordings, coverslips were transferred to a recording chamber (RC-26G; Warner Instruments, Holliston, MA, USA) fitted to the stage of an upright microscope (Axiovert 200; Zeiss) and superfused with the extracellular solution. Physiological extracellular buffer contained 140 mM NaCl, 5 mM KCl, 2 mM CaCl_2_, 2 mM MgCl_2_ and 10 mM HEPES, adjusted to pH 7.4 with NaOH. Na^+^ free extracellular buffer contains 140 mM choline chloride, 5 mM KCl, 2 mM CaCl_2_, 2 mM MgCl_2_ and 10 mM HEPES, adjusted to pH 7.4 with NaOH. Ca^2+^-free extracellular buffer for transfected HEK293 cells contained 140 mM NaCl, 5 mM KCl, 4 mM MgCl_2_, and 10 mM HEPES, adjusted to pH 7.4 with NaOH. The pipette solution, contained 140 mM KCl, 2 mM MgCl_2_, 5 mM EGTA, and 10 mM HEPES, adjusted to pH 7.4 with KOH [37,39]. Recordings were conducted at room temperature with an EPC 9 patch-clamp amplifier combined with Patchmaster software (HEKA Electronics, Reutlingen, Germany). Currents were filtered at 5 kHz and sampled at 20 kHz. Offline analyses were performed using the Fitmaster software (version 2x91, HEKA Electronics, Lambrecht, Rheinland-Pfalz, Germany). The holding potential was −70 mV, and I_K_ was evoked by 500 ms voltage steps ranging from −120 to +120 mV in 20 mV increments. Patch microelectrodes were fabricated with a Flaming/Brown micropipette puller (Sutter Instruments) and had a pipette resistance of 5–7 MΩ. Prior to recordings of DRG neurons, cells were incubated with the extracellular solution containing 10 µg/mL Alexa Fluor 488-conjugated *Griffonia simplicifolia* IB4 (#121411, Thermo Fisher Scientific) for 10 min, and the IB4-stained neurons were identified by epifluorescence illumination. AITC stock solution (5 µL, 24 mM in 20% DMSO) was added with a pipette to the bath chamber (volume 600 µL) to reach a final concentration of 200 µM, and recordings were started 60 s thereafter. A-967079 stock solution (5 µL, 1.2 mM in 20% DMSO) was added with a pipette to the bath chamber to reach a final concentration of 10 µM, and recordings were started 180 s thereafter. All recordings were taken while the superfusion system was stopped.

### 2.7. Statistics

Data are shown as mean ± SEM. Statistical evaluation was performed with GraphPad Prism 8 for Windows (GraphPad, San Diego, CA, USA). Normal distribution of data was investigated using the Kolmogorov–Smirnov test. For behavioral experiments with time courses, a two-way analysis of variance (ANOVA), followed by Sidak’s multiple comparisons test, was performed to measure effects across time between groups. An unpaired t-test was used to compare the mean of two groups in behavioral experiments, qRT-PCR analyses, calcium imaging experiments, and the quantification of cell populations. For patch clamp experiments, a two-way ANOVA with Sidak’s multiple comparisons was used for current voltage relationships between before and after AITC. No statistical methods were used to predetermine sample sizes, but the sample sizes are based on our previous knowledge and similar to standard practices in the field. No animals or data points were excluded from the analyses. The numbers of the experiments and statistical results are provided in the figure legends. A probability value of *p* < 0.05 was considered statistically significant.

## 3. Results

### 3.1. Slack^−/−^ Mice Demonstrate Increased Nociceptive Responses to TRPA1 Activation

To assess the potential role of Slack in TRPA1-mediated pain processing, we first compared the nociceptive behavior of wildtype (WT) and Slack^−/−^ mice after intraplantar injection of the TRPA1 activator AITC (10 mM in 20 µL saline containing 2% DMSO) into a hind paw. Both genotypes displayed an immediate licking and biting behavior indicative of pain, which was recorded in 5 min intervals for 30 min (Figure 1A, left). Interestingly, the cumulative paw licking and biting time over 30 min was considerably higher in Slack^−/−^ mice than in WT mice (Figure 1A, right), suggesting that Slack^−/−^ mice experience increased acute nociceptive pain after TRPA1 activation.

In addition to the immediate nocifensive behavior, intraplantar injection of AITC can also elicit a persistent hypersensitivity of the affected paw [7]. We therefore assessed the mechanical hyperalgesia using von Frey filaments during 1–72 h after AITC injection. Notably, the extent of mechanical hyperalgesia in Slack^−/−^ mice was significantly increased as compared to WT mice (Figure 1B). Control experiments using qRT-PCR revealed that TRPA1 mRNA expression is similar in DRG of WT and Slack^−/−^ mice (Figure 1C), suggesting that there was no compensatory regulation of TRPA1 due to the Slack knockout that might have contributed to the observed behavior. Together, these data suggest that Slack plays an inhibitory role in the immediate and persistent behavioral responses to TRPA1 activation.

We next assessed the nocifensive response of WT and Slack^−/−^ mice after intraplantar injection of the TRPV1 activator capsaicin (5 µg in 20 µL saline containing 2% DMSO) into a hind paw. Unlike AITC, capsaicin evoked a paw licking and biting behavior that was indistinguishable between WT and Slack^−/−^ mice during a 10 min observation period (Figure 1D). We further tested capsaicin-induced mechanical pain sensitivity but did not observe significant differences between WT and Slack^−/−^ mice (Figure 1E). qRT-PCR experiments demonstrated similar TRPV1 mRNA levels in DRGs of Slack^−/−^ and WT mice (Figure 1F; *p* = 0.487; *n* = 3). These results point to a limited contribution of Slack to TRPV1-mediated pain processing.

We then analyzed the extent of co-expression of Slack with TRPA1 and TRPV1 in sensory neurons of lumbar (L4–L5) DRG. Notably, using double-labeling in situ hybridization of Slack mRNA and TRPA1 mRNA, we found that 70.1 ± 0.3% of Slack^+^ sensory neurons expressed TRPA1, whereas 77.4 ± 3.7% of TRPA1^+^ sensory neurons expressed Slack (Figure 1G). Immunostaining using previously validated antibodies [37,39,50] revealed that only 13.9 ± 1.6% of Slack^+^ cells were positive for TRPV1, and only 16.6 ± 2.3% of TRPV1^+^ cells were positive for Slack (Figure 1H). In accordance with these findings, in a published scRNA-seq dataset of mouse DRG neurons [32], the expression of TRPA1 was highest in the NP1 cell population that also shows high *Kcnt1* (Slack) expression, whereas the expression of TRPV1 was highest in the NP3 cell population with only moderate Slack expression (Figure 1I). Hence, the observation that Slack in sensory neurons is highly co-expressed with TRPA1 but shows only minor co-expression with TRPV1 supports our behavioral findings described above. Collectively, these results indicate that Slack regulates TRPA1-mediated pain processing.

### 3.2. TRPA1-Mediated Nociceptive Responses Are Increased in Sensory Neuron-Specific Slack Mutants

The altered AITC-induced pain behavior in global Slack knockouts, in combination with the profound co-expression of Slack and TRPA1 in sensory neurons, led us to hypothesize a specific contribution of Slack in sensory neurons to AITC-induced pain processing. To test this hypothesis, we generated SNS-Slack^−/−^ mice that specifically lack Slack in Na_V_1.8^+^ sensory neurons [37,51]. The qRT-PCR analysis confirmed that Slack mRNA levels are massively reduced in DRGs of SNS-Slack^−/−^ mice but not altered in the spinal cord, cerebellum, and prefrontal cortex (Figure 2A). The proportions of sensory neuron subpopulations positive for IB4 (a marker of nonpeptidergic C fiber neurons), CGRP (peptidergic C fiber neurons), TH (C fiber low-threshold mechanoreceptor neurons), or NF200 (myelinated neurons) were similar between SNS-Slack^−/−^ and littermate control mice (Figure 2B,C). Moreover, the gross distribution of central endings of primary afferent neurons in the dorsal horn was comparable between SNS-Slack^−/−^ and littermate control mice (Figure 2D). These histological data suggest that the lack of Slack in SNS-Slack^−/−^ mice did not affect the morphology or general structural properties of sensory neurons. Moreover, control qRT-PCR experiments revealed that TRPA1 mRNA levels in DRGs were unaltered in SNS-Slack^−/−^ mice (Figure 2E).

We then analyzed the nocifensive behavior and mechanical hypersensitivity after intraplantar injection of AITC (5 µg in 20 µL saline containing 0.05% DMSO) in SNS-Slack^−/−^ and littermate control mice. Interestingly, SNS-Slack^−/−^ mice demonstrated significantly increased paw licking and biting behavior (Figure 2F). Moreover, the extent of AITC-induced mechanical hypersensitivity in SNS-Slack^−/−^ mice was increased after AITC injection (Figure 2G). Taken together, these data further support our finding that Slack and TRPA1 functionally interact in DRG neurons.

### 3.3. TRPA1-Mediated Nociceptive Responses Are Normal in Spinal Dorsal Horn Neuron-Specific Slack Mutants

A scRNA-seq study detected Slack in neurons of the spinal dorsal horn [52], suggesting that Slack might contribute to pain processing in spinal circuits. To investigate the specific contribution of Slack in dorsal horn neurons to AITC-induced nociception, we generated Lbx1-Slack^−/−^ mice which specifically lack Slack in spinal dorsal horn neurons. Quantitative RT-PCR analysis confirmed that Slack mRNA levels are significantly reduced in spinal cord extracts of Lbx1-Slack^−/−^ mice but not altered in extracts of DRGs, cerebellum, and prefrontal cortex (Figure 3A). Moreover, the distribution of inhibitory interneurons that express GAD67 [53] and excitatory interneurons positive for PKCγ [54] in the superficial dorsal horn appeared normal in Lbx1-Slack^−/−^ mice (Figure 3B), suggesting that the lack of Slack in dorsal horn neurons did not affect the morphology or general structural properties of the spinal cord. We then analyzed the behavioral responses of Lbx1-Slack^−/−^ mice to intraplantar injection of AITC (10 mM in 20 µL saline containing 2% DMSO). In contrast to Slack^−/−^ and SNS-Slack^−/−^ mice, Lbx1-Slack^−/−^ mice did not show any significant deficits in AITC-induced acute nocifensive behavior (Figure 3C), suggesting a limited contribution of Slack in dorsal horn neurons to TRPA1-mediated nociception.

### 3.4. TRPA1-Dependent Calcium Transients in Sensory Neurons Are Unaltered in Slack^−/−^ Mice

To test whether the increased AITC-induced pain behavior in Slack^−/−^ and SNS-Slack^−/−^ mice might be related to altered TRPA1-dependent Ca^2+^ influx, we compared the AITC-induced changes in intracellular Ca^2+^ in cultured DRG neurons of naïve WT and Slack^−/−^ mice by calcium imaging. As shown in Figure 4A, incubation with 200 µM AITC for 15 s increased intracellular Ca^2+^ in sensory neurons of WT and Slack^−/−^ mice to a similar extent. Both the average values of peak amplitudes (Figure 4B) and the percentage of responsive neurons (Figure 4C) were indistinguishable between groups. Viable neurons were identified by eliciting depolarization with 75 mM KCl for 20 s after AITC incubation (Figure 4A). The peak amplitudes induced by KCl were also not different in DRG neurons from WT and Slack^−/−^ mice (data not shown). We conclude that the TRPA1-dependent Ca^2+^ influx in DRG neurons is not modulated by Slack.

### 3.5. TRPA1 Activation Alters Slack-Mediated Potassium Currents in Sensory Neurons

We next investigated whether Slack-dependent potassium currents in sensory neurons were altered in response to TRPA1 activation. For that purpose, we measured total outward potassium currents (I_K_) in dissociated sensory neurons of WT and Slack^−/−^ mice in the presence of AITC by using a whole-cell patch clamp. We only analyzed IB4-positive (IB4^+^) sensory neurons, as Slack is mainly expressed in this cell population [37]. Recordings were performed before (baseline) and 60 s after adding AITC (final concentration 200 µM) with a pipette to the bath chamber. AITC decreased I_K_ in 16 of 18 neurons from WT mice and in 3 of 13 neurons from Slack^−/−^ mice. Notably, in sensory neurons of WT mice, the amplitude of I_K_ was significantly reduced after adding AITC, with a linear I-V relationship at positive potentials from +80 to +120 mV (Figure 5A). By contrast, the I_K_ amplitude in sensory neurons of Slack^−/−^ mice, which represents the Slack-independent component of I_K_, was significantly lower as compared to WT at baseline and not significantly altered after adding AITC (Figure 5A). The data suggest that TRPA1 activation reduces I_K_ in IB4^+^ sensory neurons of WT but not Slack^−/−^ mice.

TRPA1 can also be activated by highly depolarizing voltages [15,55,56]. To discern the proportion of TRPA1-mediated I_K_ versus Slack-mediated I_K_ in our experimental setting, we performed additional whole-cell patch-clamp recordings in IB4^+^ sensory neurons in the presence of the TRPA1 antagonist A-967079 [7]. Recordings were made at baseline and 3 min after adding A-967079 (final concentration 10 µM) to the bath solution. As shown in Figure 5B, the outward current at +120 mV was significantly reduced in sensory neurons of both WT and Slack^−/−^ mice in the presence of A-967079, suggesting that in our experimental setting, a proportion of I_K_ is mediated via TRPA1. We then performed control experiments in a Na^+^ free extracellular buffer (to prevent Na^+^-dependent activation of Slack) and observed that AITC applied to the bath solution did not affect I_K_ in sensory neurons from both WT and Slack^−/−^ mice (Figure 5C).

The observation that in a physiological extracellular buffer, AITC *reduced* the I_K_ in sensory neurons of WT mice was unexpected because based on the exaggerated AITC-evoked pain behavior in Slack^−/−^ and SNS-Slack^−/−^ mice, we hypothesized that Slack-mediated I_K_ would be increased after TRPA1 activation in sensory neurons. Given the fact that (i) TRPA1 activation may induce substantial Ca^2+^ influx, and this Ca^2+^ permeation restricts monovalent cation (such as Na^+^) flux through TRPA1 [21,57,58], and (ii) Slack is not only activated by intracellular Na^+^ but inhibited by intracellular Ca^2+^ [39,59], we reasoned that the observed reduction in I_K_ in the presence of AITC in a physiological extracellular buffer might be driven by Ca^2+^ influx in our patch-clamp setting, which contained 2 mM Ca^2+^ in the external solution. However, recordings in sensory neurons with a Ca^2+^-free external solution are not feasible because Ca^2+^ has a dramatic effect on cell mentalism [58], and therefore, sufficient amounts of Ca^2+^ in the external solution are required [60]. Nevertheless, our data show that the activity of Slack channels in sensory neurons can be modulated in a TRPA1-dependent manner.

### 3.6. TRPA1 Activation Increases Slack-Mediated Potassium Currents In Vitro

To further characterize the interaction between Slack and TRPA1, we used a HEK-293 cell line stably expressing human Slack (herein referred to as HEK-Slack cells) and transiently transfected GFP-tagged human TRPA1 into these cells. This strategy led to the expression of TRPA1 in about 35% of transfected cells (as indicated by green fluorescence in microscopy; the resulting TRPA1-positive cells were referred to as HEK-Slack-TRPA1 cells). A series of whole-cell patch-clamp recording experiments were conducted using a Ca^2+^-free extracellular buffer in order to avoid Ca^2+^-mediated inhibition of Slack activity [39,59]. As shown in Figure 6, the addition of AITC (200 µM) to the external solution did not affect I_K_ in HEK-Slack cells. However, in HEK-Slack-TRPA1 cells, I_K_ was considerably increased in the presence of AITC at positive potentials (+60 mV to 120 mV) (Figure 6B). Furthermore, the application of TRPA1 antagonist A-967079 reduced IK in HEK-Slack-TRPA1 cells (Figure 6C). Interestingly, when the Ca^2+^-free extracellular buffer was replaced by the physiological extracellular buffer, the AITC-evoked increase in I_K_ was substantially ameliorated in HEK-Slack-TRPA1 cells (Figure 6D; compare to Figure 6B). Altogether, these results further support our finding that TRPA1 may functionally interact with Slack.

## 4. Discussion

Using Slack^−/−^, SNS-Slack^−/−^, and Lbx1-Slack^−/−^ mouse strains, we here provide evidence that Slack controls TRPA1-triggered pain. The high degree of colocalization of Slack with TRPA1, but not TRPV1, is in accordance with a specific functional coupling of TRPA1 and Slack in sensory neurons. Our electrophysiological experiments further suggest that Slack-mediated potassium currents can be modulated by TRPA1 activation.

The cellular distribution of TRPA1 in sensory neurons has been investigated in several previous studies. However, the detailed TRPA1 expression pattern still remains controversial. For example, in early in situ hybridization experiments, TRPA1 has been nearly exclusively detected in TRPV1^+^ sensory neurons of adult mice, and double-labeling suggested that 30% of TRPV1^+^ neurons express TRPA1 [22]. Using immunohistochemistry, another study confirmed an exclusive expression of TRPA1 in TRPV1^+^ neurons and detected partial co-expression of TRPA1 and CGRP, a marker of peptidergic C-fiber neurons [25]. However, other studies found more TRPA1 in IB4-binding, non-peptidergic sensory neurons [35,36,61], of which only a minority express CGRP or TRPV1 in adulthood [62]. Discrepancies between studies may have resulted from the limitations of in situ hybridization and immunohistochemistry techniques [36]. It should be also considered that a TRPA1 splice variant has been identified that may interact with full-length TRPA1 and alter its expression at the plasma membrane, thereby theoretically affecting the results of tissue staining experiments [63]. Our observation that the majority of Slack^+^ sensory neurons (70%) co-express TRPA1, but only 14% are positive for TRPV1, provides indirect evidence for only partial overlap of TRPA1 and TRPV1 in sensory neurons. This finding is consistent with recent scRNA-seq studies [32,33]. Hence, the distribution pattern of Slack, TRPA1, and TRPV1 in sensory neurons that we report in our study further supports the finding that Slack modulates TRPA1-induced, but not TRPV1-induced, pain processing.

Previous studies in our lab and by others found that Slack significantly contributed to neuropathic pain [37,39,64] and acute itch [38]. However, Slack^−/−^ mice showed normal sensitivities to heat or cold stimuli and unaltered hypersensitivity in inflammatory pain models [37]. Furthermore, we previously observed that 0.5% formalin-induced nociceptive behavior is normal in Slack^−/−^ mice [37]. As the formalin-induced nociceptive behavior significantly depends on TRPA1 [19], it was surprising that AITC-mediated nociceptive behavior was altered in Slack^−/−^ and SNS-Slack^−/−^ mice. These apparently opposing findings may reflect the complex mechanisms underlying nociception in vivo. Although speculative, the difference could be due to the fact that AITC is a more specific TRPA1 agonist as compared to formalin [15,65]. It should be noted that in our AITC behavior experiments, 2% and 0.05% DMSO were used as vehicles in Slack^−/−^ and SNS-Slack^−/−^ mice, respectively. Previous reports that used a higher concentration of DMSO (20–25%) for intraplantar injection did not find a vehicle effect [66,67]. Hence, we do not expect that the pain behavior in our experiments was affected by different concentrations of DMSO. It is also worth noting that the alteration in AITC-induced mechanical hypersensitivity was more pronounced in Slack^−/−^ mice (significantly different from WT mice 3 h to 48 h after the AITC injection) than in SNS-Slack^−/−^ mice (significantly different from control mice only at 24 h after the AITC injection). It therefore seems possible that Slack channels expressed in Na_V_1.8-negative sensory neurons (in which Slack is not knocked out in SNS-Slack^−/−^ mice) or in the central nervous system might also contribute to TRPA1-mediated pain processing.

Based on our findings that TRPA1-mediated nociceptive behavior is increased in Slack^−/−^ and SNS-Slack^−/−^ mice, the question arises of how TRPA1 functionally interacts with Slack in sensory neurons. In general, Slack is activated by elevations in intracellular Na^+^, driven by voltage-dependent Na^+^ channels, N-methyl-D-aspartic acid receptors or other nonselective cation channels, and responsible for a delayed outward current termed I_KNa_ [68]. On the other hand, its activity is inhibited by intracellular divalent cations that modify channel gating by an allosteric mechanism [39,59,69] Furthermore, Slack activity is regulated by a variety of signaling pathways, including phosphorylation of the C-terminal domain by protein kinase C [70,71], transmembrane protein TMEM16C [72], G protein-coupled receptors [73], and the fragile X mental retardation protein [74,75,76]. Although highly speculative, a TRPA1-driven influx of Na^+^ might lead to Slack activation and subsequent K^+^ efflux that limits the nociceptor activity in vivo. However, in our patch-clamp recordings in sensory neurons, we did observe a reduced Slack-mediated I_K_ after TRPA1 activation. In the dilated state, the permeability sequence through TRPA1 has been calculated to be Ca^2+^ > Ba^2+^ > Mg^2+^ > NH_4_^+^ > Li^+^ > Na^+^ > K^+^ > Rb^+^, suggesting that binding of calcium in the pore may effectively hinter monovalent cation permeation [57,58]. Therefore, we speculate that in the experimental setting of patch-clamp analyses in sensory neurons with 2 mM Ca^2+^ in the external solution, the Slack activity is inhibited. The increased I_K_ after TRPA1 activation in HEK-Slack-TRPA1 cells in a Ca^2+^-free setting further supports our hypothesis. However, we cannot exclude the possibility that Slack activity is modulated by other mechanisms upon TRPA1 activation. Further studies are required to determine how exactly TRPA1 activation affects Slack I_K_ currents in sensory neurons.

## 5. Conclusions

Overall, our findings suggest that Slack in sensory neurons limits the AITC-induced pain processing. These data provide further insights into the molecular mechanisms of TRPA1-mediated nociception.

## Figures and Tables

**Figure 1 cells-11-01693-f001:**
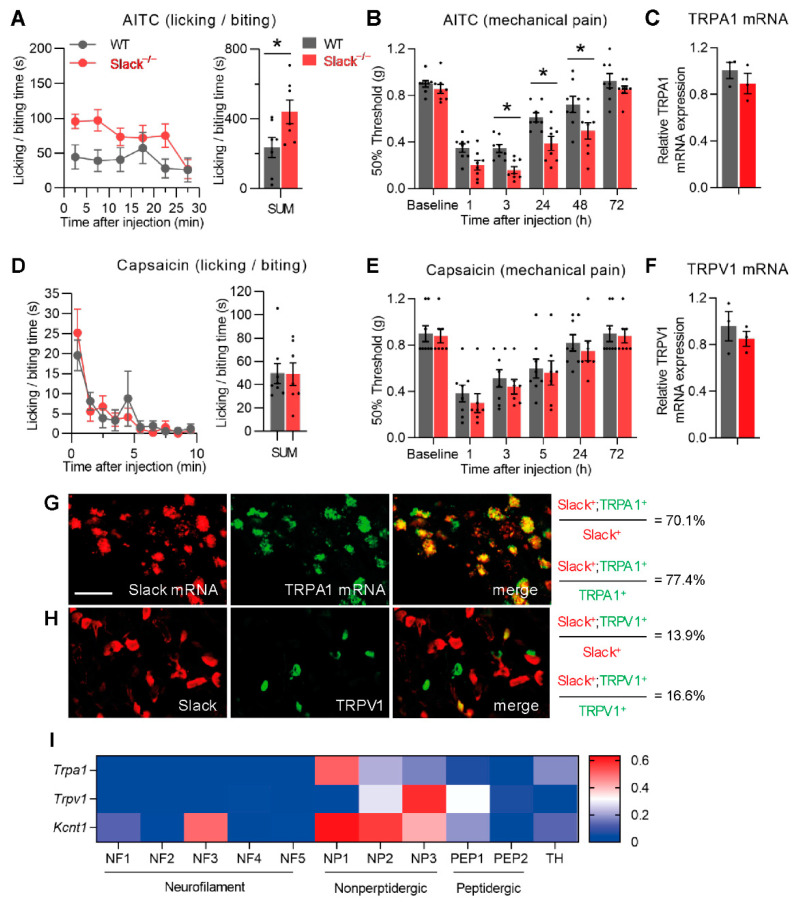
Slack^−/−^ mice display increased allyl isothiocyanate (AITC)-evoked but normal capsaicin-evoked pain behavior. (**A**) Time course of paw licking and biting (left) and the sum of licking and biting time over 30 min (right; *p* = 0.0394; *n* = 7 mice per group) after intraplantar AITC injection in wildtype (WT) and Slack^−/−^ littermates. (**B**) Time course of mechanical hypersensitivity after intraplantar AITC injection. Two-way analysis of variance (ANOVA), effect of genotype (*p* = 0.0007) with Sidak’s multiple comparison test (*p* values represent comparisons between genotypes for each time point: 3 h, *p* = 0.0421; 24 h, *p* = 0.0076; 48 h, *p* = 0.0064); *n* = 8 mice per group. Note that both licking/biting and mechanical hypersensitivity are significantly increased in Slack^−/−^ mice after AITC injection. (**C**) Quantitative RT-PCR in DRGs of WT and Slack^−/−^ mice revealed that the transient receptor potential (TRP) ankyrin 1 (TRPA1) mRNA expression is not compensatorily regulated in the absence of Slack (*p* = 0.3727; *n* = 3 mice per group). (**D**) Time course of paw licking and biting (left) and the sum of licking and biting time over 10 min (right; *p* = 0.9621; *n* = 7–8 mice per group) after intraplantar capsaicin injection. (**E**) Time course of mechanical hypersensitivity after intraplantar capsaicin injection. Two-way ANOVA, effect of genotype (*p* = 0.5893; *n* = 7–8 mice per group). Note that the capsaicin-induced pain behavior was unaltered in Slack^−/−^ mice. (**F**) Quantitative RT-PCR in DRGs of WT and Slack^−/−^ mice revealed that TRP vanilloid 1 (TRPV1) mRNA expression is not compensatory regulated in the absence of Slack (*p* = 0.4874; *n* = 3 mice per group) (**G**) Double in situ hybridization of Slack mRNA and TRPA1 mRNA in DRGs. Scale bar, 50 µm. (**H**) Double-labeling immunostaining of Slack and TRPV1 in DRGs. A quantitative summary of co-expression in G (180 Slack-positive cells from 3 mice were counted) and H (204 Slack-positive cells from 4 mice were counted) is shown on the right. (**I**) Expression of TRPA1, TRPV1, and Slack (gene *Kcnt1*) across sensory neuron subsets from published scRNA-seq data. Data are presented as mean ± SEM. * *p* < 0.05.

**Figure 2 cells-11-01693-f002:**
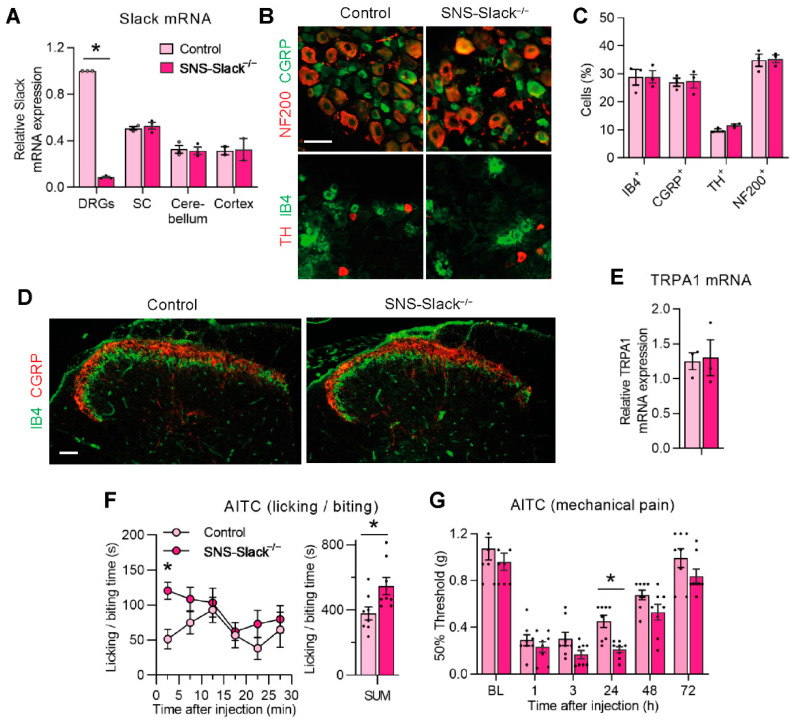
SNS-Slack^−/−^ mice display increased AITC-evoked pain behavior. (**A**) Quantitative RT-PCR in lumbar DRGs, lumbar spinal cord, cerebellum and prefrontal cortex revealed that Slack mRNA is selectively reduced in DRGs of SNS-Slack^−/−^ mice (*p* = 0.0002; *n* = 3 mice per group). (**B**,**C**) Expression pattern and percentages of DRG neurons binding IB4 (2207 cells from 4 mice per group were counted), or immunoreactive for CGRP (2110 cells from 4 mice per group were counted), TH (839 cells from 4 mice per group were counted), NF200 (2751 cells from 4 mice per group were counted), are similar in SNS-Slack^−/−^ and control mice. Scale bar, 50 µm. (**D**) The distribution of central terminals of primary afferents immunoreactive for CGRP or binding IB4 in the dorsal horn appears normal in SNS-Slack^−/−^ mice. Scale bar, 50 µm. (**E**) Quantitative RT-PCR revealed that TRPA1 mRNA expression in lumbar DRGs is similar in control and SNS-Slack^−/−^ mice (*p* = 0.8638; *n* = 3 mice per group). (**F**) Time course of paw licking and biting (left; *p* = 0.0148 for the 0–5 min period) and the sum of licking and biting time over 30 min (right; *p* = 0.0238; *n* = 8 mice per group) after intraplantar injection of AITC in control and SNS-Slack^−/−^ littermates. (**G**) Time course of mechanical hypersensitivity after intraplantar AITC injection. Two-way ANOVA, effect of genotype (*p* = 0.0126) with Sidak’s multiple comparisons test (*p* = 0.0346, representing comparisons between genotypes for the 24 h time point); *n* = 8 mice per group. Data are presented as mean ± SEM. * *p* < 0.05.

**Figure 3 cells-11-01693-f003:**
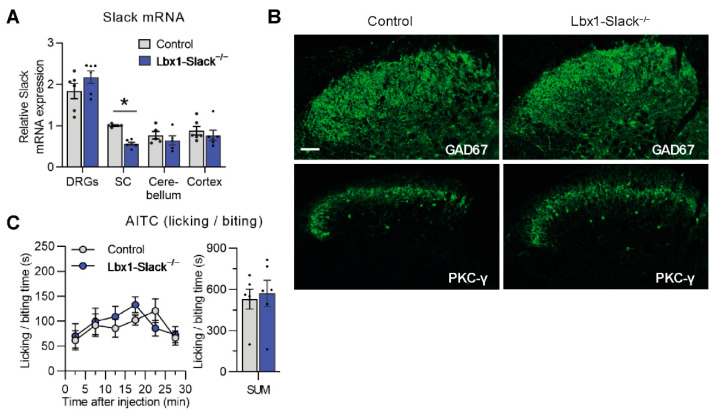
Lbx1-Slack^−/−^ mice display normal AITC-evoked pain behavior. (**A**) Quantitative RT-PCR in lumbar DRGs, lumbar spinal cord, cerebellum, and prefrontal cortex revealed that Slack mRNA levels are selectively reduced in the spinal cord of Lbx1-Slack^−/−^ mice (*p* < 0.0001; *n* = 6 mice per group). (**B**) The distribution of GAD67^+^ inhibitory interneurons and PKCγ^+^ excitatory interneurons in the dorsal horn appears normal in Lbx1-Slack^−/−^ mice. Scale bar, 50 µm. (**C**) Time course of paw licking and biting (left) and the sum of licking and biting time over 30 min (right; *p* = 0.7319; *n* = 6 mice per group) after intraplantar AITC injection in Lbx1-Slack^−/−^ and control littermates. Data are presented as mean ± SEM. * *p* < 0.05.

**Figure 4 cells-11-01693-f004:**
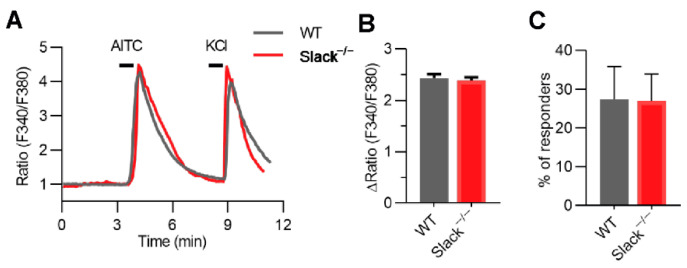
AITC-mediated calcium influx is normal in sensory neurons of Slack^−/−^ mice. (**A**) Representative examples of Fura-2-ratiometric calcium traces evoked by AITC and KCl in cultured lumbar DRG neurons of WT and Slack^−/−^ mice. (**B**) Magnitude of the calcium response to AITC stimulation (WT, *n* = 452 neurons in 3 mice; Slack^−/−^, *n* = 476 neurons in 3 mice; *p* = 0.7000). (**C**) Percentage of responsive neurons to AITC stimulation (*p* = 0.9739). These data show that AITC-evoked calcium responses are normal in DRG neurons from Slack^−/−^ mice. Data in B and C are presented as mean ± SEM.

**Figure 5 cells-11-01693-f005:**
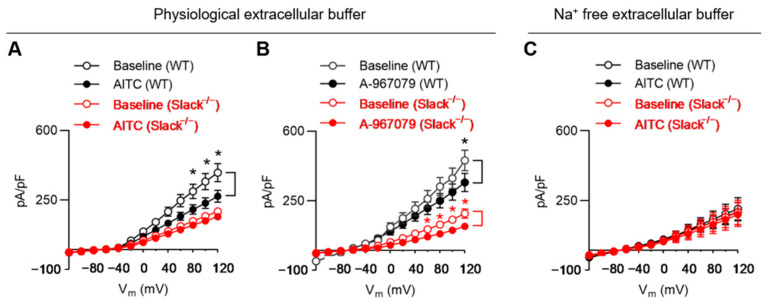
AITC-mediated modulation of potassium currents in IB4^+^ sensory neurons from WT and Slack^−/−^ mice. (**A**) IV relations of outward potassium currents (I_K_) obtained in whole-cell patch-clamp recordings in IB4^+^ sensory neurons from 4 WT (*n* = 18 cells) and 4 Slack^−/−^ mice (*n* = 13 cells) before and after AITC (200 µM) application in the physiological extracellular buffer. Note that in this experimental setting (which includes 2 mM Ca^2+^ and 140 mM Na^+^ in the external solution), TRPA1 activation led to a significant reduction in I_K_ in sensory neurons from WT but not Slack^−/−^ mice. (**B**) IV relations of I_K_ in sensory neurons from 3 WT (*n* = 13 cells) and 3 Slack^−/−^ mice (*n* = 12 cells) before and after application of the TRPA1 antagonist A-967079 (10 µM) in the physiological extracellular buffer. The TRPA1 antagonist significantly reduced I_K_ in sensory neurons from both WT and Slack^−/−^ mice. (**C**) IV relations of I_K_ in sensory neurons from 3 WT (*n* = 9 cells) and 3 Slack^−/−^ mice (*n* = 8 cells) before and after AITC (200 µM) application in a Na^+^ free extracellular buffer. In this experimental setting (which includes 2 mM Ca^2+^ but no Na^+^ in the external solution), TRPA1 activation did not alter I_K_ in sensory neurons from both WT and Slack^−/−^ mice. Data are presented as mean ± SEM. * *p* < 0.05.

**Figure 6 cells-11-01693-f006:**
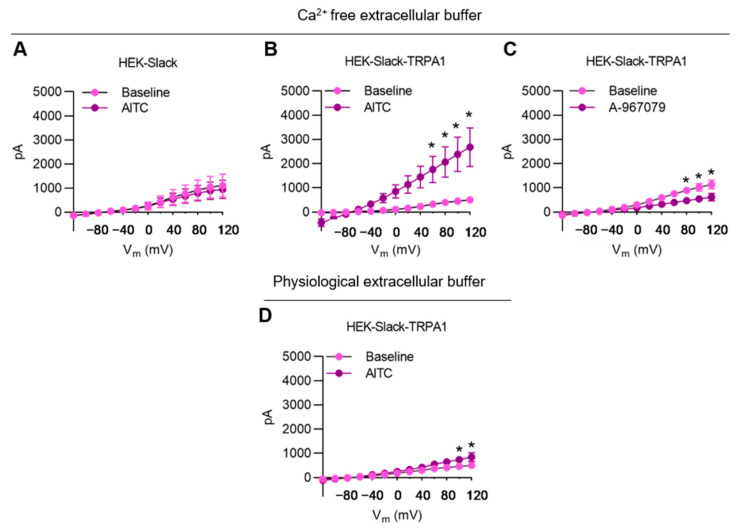
AITC-mediated modulation of potassium currents in transfected HEK293 cells. (**A**,**B**) IV relations of I_K_ in HEK-Slack cells ((**A**); *n* = 5 cells) and HEK-Slack-TRPA1 cells ((**B**); *n* = 14 cells) obtained in whole-cell patch-clamp recordings using a Ca^2+^-free external solution before and after TRPA1 activation by 200 µM AITC. (**C**) IV relations of I_K_ in HEK-Slack-TRPA1 cells before and after TRPA1 inhibition by 10 µM A-967079 (*n* = 9 cells) in whole-cell patch-clamp recordings using a Ca^2+^-free external solution. (**D**) IV relations of I_K_ in HEK-Slack-TRPA1 cells before and after 200 µM AITC application (*n* = 9 cells) in whole-cell patch-clamp recordings in physiological extracellular buffer Note that in a Ca^2+^-free external solution, Slack-mediated I_K_ is increased after TRPA1 activation. Data are presented as mean ± SEM. Paired *t* test, * *p* < 0.05.

## Data Availability

Not applicable.

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
