# Peer review of "Slack Potassium Channels Modulate TRPA1-Mediated Nociception in Sensory Neurons"

_cells, 2022, doi:10.3390/cells11101693_

Round 1
Reviewer 1 Report
In this paper, the authors explore the role of the potassium channel Slack in the TRPA1-mediated activation of sensory neurons. They demonstrate a profound co-expression of Slack and TRPA1 in a specific subset of sensory neurons and, using various knockout mouse strains, they provide evidence that Slack regulates TRPA1-triggered peripheral pain.
Overall, this is a straightforward study, carefully conducted and written in a concise style. The experiments are logically sequenced and performed with evident competence. The behavioral experiments are interesting and may form the basis of further studies.
I have several comments regarding the experimental design of patch-clamp experiments and several minor suggestions below.
Major:
The design of the experiments shown in Figure 5 and 6 seems not very well thought out and I wonder if they are necessary for the basic message of the paper:
- The current after AITC decreased approximately by 20% in Figure 5A and by 14% in Figure B. That's really not much of a difference (although the representative recording shown in Fig. 5C shows a decrease by 54%). Although the paired t-test showed a significant difference in Fig. 5A, I'd like to see the original raw data on which the significant difference was determined.
- TRPA1 is a voltage-activated channel and, under the conditions used by the authors, it should exhibit substantial activity at +120mV (the reported half-maximal activation voltage is about +90 to +120 mV). Thus, IK current in IB+ neurons should be not only Slack-mediated but also TRPA1 mediated (Line 412). However, the proportion of Slack and TRPA1- mediated outward currents is not clear.
- In particular, the recordings shown in Fig. 5-6 were performed with AITC that was (Lines 235-6:) “added with a pipette to the bath chamber”. That is, the authors had no control over how AITC diffuses and acts in a time-dependent manner on a cell. The authors do not mention whether the cells were verified to be sensitive to AITC, which is a crucial parameter that should be correlated with the observed effects. Generally, TRPA1 activation is multiphasic and after maximum activation by high concentration of AITC, the channels desensitize. The time at which the current was recorded after the AITC was added is not indicated. The difference between the current densities in WT and Slack-/- cells indicates that Slack could positively modulate TRPA1 at depolarizing potentials, but not at a resting potential (Figure 4).
- Line 403: The authors state that only IB4-positive sensory neurons were analyzed. How were the cells tested for IB4-positivity in these experiments?
- The experiments demonstrated in Figure 6 can be interpreted as indicating that the presence of AITC sensitized TRPA1-mediated currents evoked by depolarization. The only proof here would be to show that the currents are not blocked by a TRPA1 inhibitor, or using the same protocol to compare the currents with HEKs that do not express Slack. Otherwise, the IK outward currents could be composed of Slack- and TRPA1-mediated currents. In both cases, Figure 5 and Figure 6, the time course of the AITC activation should be shown, or specific inhibitors of TRPA1 or Slack should be used to discern Slack-mediated component of the currents.
Minor:
- Line 4, should be “Sisignano” instead of “Sisignanao”
- Lines 14-15, should be “….is highly expressed in a subset of sensory neurons where it acts as an essential…”
- Line 55, should be “TRPA1” instead of “TPRA1”
- Line 249, p < 0.05 instead of P < 0.05 (or edit „p“ throughout the text)
- Line 261, should be „allyl isothiocyanate“ instead of „ally lisothiocyanate“
- Lines 265-266, „comparisons“ instead of „comparisions“
- Line 270, should be „compensatorily“ instead of „compensatory“
- Lines 325-326, Figure 2B and C instead of Figure 2B+C
- Line 351, „comparisons“ instead of „comparisions“
- Line 371, caption in Figure 3, panel C, „AITC (licking“ instead of „AITC (lcking“
- Line 393, Fura-2-ratiometric instead of Fura-2-ratiometirc
- Line 409, Figure 5C instead of Figure 5B
- Line 411, Figure 5B instead of Figure 5 (but the panels should be referenced in the order A,B,C in the text)
Author Response
In answer to reviewer #1:
In this paper, the authors explore the role of the potassium channel Slack in the TRPA1-mediated activation of sensory neurons. They demonstrate a profound co-expression of Slack and TRPA1 in a specific subset of sensory neurons and, using various knockout mouse strains, they provide evidence that Slack regulates TRPA1-triggered peripheral pain.
Overall, this is a straightforward study, carefully conducted and written in a concise style. The experiments are logically sequenced and performed with evident competence. The behavioral experiments are interesting and may form the basis of further studies.
I have several comments regarding the experimental design of patch-clamp experiments and several minor suggestions below.
Major:
The design of the experiments shown in Figure 5 and 6 seems not very well thought out and I wonder if they are necessary for the basic message of the paper:
The current after AITC decreased approximately by 20% in Figure 5A and by 14% in Figure B. That's really not much of a difference (although the representative recording shown in Fig. 5C shows a decrease by 54%). Although the paired t-test showed a significant difference in Fig. 5A, I'd like to see the original raw data on which the significant difference was determined.
We have performed additional patch-clamp experiments in order to increase the number of neurons per group. Moreover, we changed the statistical method from paired t-test to 2way-ANOVA with Sidak´s multiple comparisons. These results are shown in the new Figure 5A and B. In neurons from WT mice, significant differences before versus after AITC are observed at +80, +100, and +120 mV (Fig. 5A).
TRPA1 is a voltage-activated channel and, under the conditions used by the authors, it should exhibit substantial activity at +120mV (the reported half-maximal activation voltage is about +90 to +120 mV). Thus, IK current in IB+ neurons should be not only Slack-mediated but also TRPA1 mediated (Line 412). However, the proportion of Slack and TRPA1- mediated outward currents is not clear.
We agree with the reviewer that in our protocol the IK current might be not only Slack-mediated but also TRPA1 mediated. We therefore performed additional whole-cell patch-clamp recordings with the TRPA1 antagonist A-967079, which are presented in the new Figures 5B and 6C. We found that in presence of A-967079 the IK current was decreased in sensory neurons of both WT and Slack-/- mice. These data indicate that the IK current is not only Slack-mediated but also TRPA1 mediated, as suggested by the reviewer.
Furthermore, in control experiments with a Na+ free extracellular buffer (to prevent Na+-dependent activation of Slack) we observed that AITC applied to the bath solution did not affect IK in sensory neurons from both WT and Slack-/- mice. These data are presented as new Figure 5C.
We modified the respective sentence (former line 412) as follows (now line 465): “TRPA1 can also be activated by highly depolarizing voltages [15, 55, 56]. To discern the proportion of TRPA1-mediated IK versus Slack-mediated IK in our experimental set-ting, we performed additional whole-cell patch-clamp recordings in IB4+ sensory neurons in presence of the TRPA1 antagonist A-967079 [7]. Recordings were made at baseline and 3 min after adding A-967079 (final concentration 10 µM) to the bath solution. As shown in Figure 5B, the outward current at +120 mV was significantly reduced in sensory neurons of both WT and Slack-/- mice in presence of A-967079, suggesting that in our experimental setting a proportion of IK is mediated via TRPA1. We then performed control experiments in a Na+ free extracellular buffer (to prevent Na+-dependent activation of Slack) and observed that AITC applied to the bath solution did not affect IK in sensory neurons from both WT and Slack-/- mice (Figure 5C). ”
In particular, the recordings shown in Fig. 5-6 were performed with AITC that was (Lines 235-6:) “added with a pipette to the bath chamber”. That is, the authors had no control over how AITC diffuses and acts in a time-dependent manner on a cell.
In our patch-clamp recordings we used a RC-26G chamber (Warner Instruments, Holliston, USA) that facilitates laminar solution flow throughout the bath area. Therefore, we suppose that AITC rapidly reached all cells after adding to this bath chamber. Furthermore, all recordings were taken while the superfusion system wasn´t running.
We now included this information in the Materials and Methods section.
Line 255: “For whole-cell voltage clamp recordings, coverslips were transferred to a recording chamber (RC-26G; Warner Instruments, Holliston, USA) fitted to the stage of an upright microscope (Axiovert 200; Zeiss) and superfused with extracellular solution.”
Line 278: “All recordings were taken while the superfusion system was stopped.”
The authors do not mention whether the cells were verified to be sensitive to AITC, which is a crucial parameter that should be correlated with the observed effects.
We now indicate that cells were verified to be sensitive to AITC in line 457: “AITC decreased IK in 16 of 18 neurons from WT mice and in 3 of 13 neurons from Slack-/- mice.”
Generally, TRPA1 activation is multiphasic and after maximum activation by high concentration of AITC, the channels desensitize. The time at which the current was recorded after the AITC was added is not indicated.
Recordings were started 60 s after adding AITC to the bath chamber. We included this information in the Materials and Methods section (line 274):
“AITC stock solution (5 µl, 24 mM in 20% DMSO) was added with a pipette to the bath chamber (volume 600 µl) to reach a final concentration of 200 µM and recordings were started 60 s thereafter.”
Furthermore, we added this information to the Results section (line 455):
“Recordings were performed before (baseline) and 60 s after adding AITC (final concentration 200 µM) with a pipette to the bath chamber.”
The difference between the current densities in WT and Slack-/- cells indicates that Slack could positively modulate TRPA1 at depolarizing potentials, but not at a resting potential (Figure 4).
We agree with the conclusion of the reviewer that Slack seems to modulate TRPA1 at depolarizing potentials, but not at a resting potential.
Line 403: The authors state that only IB4-positive sensory neurons were analyzed. How were the cells tested for IB4-positivity in these experiments?
We added this information to the Materials and Methods section.
Line 271: “Prior to recordings of DRG neurons, cells were incubated with extracellular solution containing 10 µg/ml Alexa Fluor 488-conjugated Griffonia simplicifolia IB4 (#121411, Thermo Fisher Scientific) for 10 min, and the IB4-stained neurons were identified by epifluorescence illumination.”
The experiments demonstrated in Figure 6 can be interpreted as indicating that the presence of AITC sensitized TRPA1-mediated currents evoked by depolarization. The only proof here would be to show that the currents are not blocked by a TRPA1 inhibitor, or using the same protocol to compare the currents with HEKs that do not express Slack. Otherwise, the IK outward currents could be composed of Slack- and TRPA1-mediated currents. In both cases, Figure 5 and Figure 6, the time course of the AITC activation should be shown, or specific inhibitors of TRPA1 or Slack should be used to discern Slack-mediated component of the currents.
We performed additional patch-clamp recordings with the TRPA1 antagonist A-967079 in sensory neurons (Figure 5) and in HEK-Slack-TRPA1 cells (Figure 6).
Concerning the recordings with A-967079 in sensory neurons (Figure 5), please see our answer to your comment above.
In HEK-Slack-TRPA1 cells we observed that IK currents were decreased after adding A-967079 to the Ca2+ free extracellular bath solution (Figure 6C). Furthermore, we analyzed how IK currents in HEK-Slack-TRPA1 cells are affected by AITC in a physiological extracellular buffer. These recordings revealed that the AITC-evoked increase in IK was substantially ameliorated in physiological buffer as compared to Ca2+ free buffer (Figure 6D; compare to Figure 6B).
These findings are described at line 517: “However, in HEK-Slack-TRPA1 cells IK was considerably increased in presence of AITC at positive potentials (+60 mV to 120 mV) (Figure 6B). Furthermore, application of TRPA1 antagonist A-967079 reduced IK in HEK-Slack-TRPA1 cells (Figure 6C). Interestingly, when the Ca2+-free extracellular buffer was replaced by physiological extracellular buffer, the AITC-evoked increase in IK was substantially ameliorated in HEK-Slack-TRPA1 cells (Figure 6D; compare to Figure 6B).”
Minor:
Line 4, should be “Sisignano” instead of “Sisignanao”
We corrected it to “Sisignano”.
Lines 14-15, should be “….is highly expressed in a subset of sensory neurons where it acts as an essential…”
It has been corrected.
Line 55, should be “TRPA1” instead of “TPRA1”
It has been corrected.
Line 249, p < 0.05 instead of P < 0.05 (or edit „p“ throughout the text)
“p < 0.05” has been used throughout the text.
Line 261, should be „allyl isothiocyanate“ instead of „ally lisothiocyanate“
It has been corrected (now line 306).
Lines 265-266, „comparisons“ instead of „comparisions“
It has been corrected (now line 311).
Line 270, should be „compensatorily“ instead of „compensatory“
It has been corrected (now line 315).
Lines 325-326, Figure 2B and C instead of Figure 2B+C
It has been corrected (now line 372).
Line 351, „comparisons“ instead of „comparisions“
It has been corrected (now line 401).
Line 371, caption in Figure 3, panel C, „AITC (licking“ instead of „AITC (lcking“
The caption in Figure 3C has been corrected.
Line 393, Fura-2-ratiometric instead of Fura-2-ratiometirc
It has been corrected (now line 444).
Line 409, Figure 5C instead of Figure 5B
The sentence has been deleted in the revised version.
Line 411, Figure 5B instead of Figure 5 (but the panels should be referenced in the order A,B,C in the text)
Ok. It has been corrected (now Figure 5A).

Reviewer 2 Report
In this manuscript the Authors investigated whether Slack channels may modulate the activity of TRPA1+ sensory neurons.
Below are listed my comments and suggestions.
- Why pain behavior after AITC and capsaicin injection was differently recorded (30 min for AITC and 10 min for capsaicin)?
- Was mechanical sensitivity performed right after AITC/capsaicin-induced pain behavior recordings?
- Line 301: “Due to the lack of a specific TRPA1 antibody we performed double-labeling in situ hybridization of Slack mRNA and TRPA1 mRNA” What does this sentence stand for? There are a lot of studies investigating TRPA1 immunostaining in DRG/TG.
- Section 2.4: to describe the image analysis procedure. Eg: the number of slices that have been considered for each animal.
- Section 2.7: the sentence in line 245-246 is not a justification for lack of sample size calculation. If the authors refer to other publications, then they should have simply done a sample size calculation taking into consideration such studies. Moreover, only for behavioral tests n=7-8 was applied, but for the remaining evaluation only n=3 was considered. I do not believe it is sufficient. It is not indicated if the data were tested for normality (and which test was used). I do not believe n=3 should be tested with a parametric test. To clearly state the total number of animals used.
- It doesn’t seem that the authors took into account the issue of the splice variant of mouse TRPA1 (https://doi.org/10.1038/ncomms3399). Please revise and discuss this point.
- A TRPA1 antagonist should have been used to confirm the obtained data.
In the discussion section poor attention was given to the behavioral data. For instance, the data reported in Figure 1B and Figure 2G show a different pattern (in terms of time) of behavioral response after mechanical stimulation in Slack-/- and SNS-Slack-/-.
Author Response
In answer to reviewer #2:
In this manuscript the Authors investigated whether Slack channels may modulate the activity of TRPA1+ sensory neurons. Below are listed my comments and suggestions.
Why pain behavior after AITC and capsaicin injection was differently recorded (30 min for AITC and 10 min for capsaicin)?
In previous studies we observed that intraplantar injection of AITC in mice evokes a licking and biting behavior that normally persists for 15-30 min (Lu et al., Neuropharmacology 2017), whereas intraplantar injection of capsaicin evokes paw licking and biting over 3-10 min (Lu et al., Pain 2014; Lu et al., Neuropharmacology 2017). We therefore recorded the pain behavior for 30 min after AITC injection and for 10 min after capsaicin injection.
Was mechanical sensitivity performed right after AITC/capsaicin-induced pain behavior recordings?
Yes. We assessed the mechanical sensitivity after the AITC/capsaicin-induced paw licking and biting recordings. We added this information in the Materials and Methods section, lines 107 and 112:
“Immediately thereafter, mice were placed in boxes on an elevated metal mesh floor and mechanical sensitivity was evaluated over 72 h as described above.”
Line 301: “Due to the lack of a specific TRPA1 antibody we performed double-labeling in situ hybridization of Slack mRNA and TRPA1 mRNA” What does this sentence stand for? There are a lot of studies investigating TRPA1 immunostaining in DRG/TG.
We agree that TRPA1 immunostaining in DRG/TG has been performed in several studies, however with mixed results. To prevent missinterpretation, we have removed this sentence.
Section 2.4: to describe the image analysis procedure. Eg: the number of slices that have been considered for each animal.
We have now added this information in the Materials and Methods section (line 195):
“For quantification of Slack mRNA-positive DRG neuron populations, we cut serial sections of lumbar DRGs (L4-L5) from 3 mice. Per animal ≥ 2 sections at least 100 µm apart with at least 100 cells were counted manually by an observer (346 cells in total). Only cells containing DAPI-positive nuclei and showing clear staining signals above background level, with a threshold set based on scramble control hybridization, were included.
For quantification of markers of DRG neuron subpopulations, serial sections of lum-bar DRGs (L4-L5) from SNS-Slack-/- and WT mice (3-4 mice per genotype) were cut. Per animal ≥ 2 sections at least 100 µm apart with at least 100 cells were counted manually by an observer (17455 cells in total). Only cells showing clear staining signals above back-ground level were included. For calculation of the percentage of marker-positive DRG neurons, the total number of DRG neuron somata was counted based on their autofluo-rescence visualized in the FITC channel.”
Section 2.7: the sentence in line 245-246 is not a justification for lack of sample size calculation. If the authors refer to other publications, then they should have simply done a sample size calculation taking into consideration such studies.
We have modified the sentence in line 245-246 (now line 289) as follows: “No statistical methods were used to predetermine sample sizes, but the sample sizes are based on our previous knowledge and similar standard practices in the field.”
Moreover, only for behavioral tests n=7-8 was applied, but for the remaining evaluation only n=3 was considered. I do not believe it is sufficient.
In our behavioral studies with paw injection of compounds, we usually investigate 6-8 animals per group. By contrast, for quantitative RT-PCR analyses we normally include tissues from 3-4 animals per group and measure the samples in duplicate or triplicate (Flauaus et al., Anesthesiology 2022; Petersen et al., Pain 2019; Kallenborn-Gerhardt et al., Pain 2017; Lu et al., J Neurosci 2015; Lu et al., Pain 2014; Kallenborn-Gerhardt et al., J Neurosci 2012; Heine et al., J Neurosci 2011; Schmidtko et al., J Neurosci 2008). To our experience, samples from 3-4 animals per group is sufficient for RT-PCR experiments in our studies.
It is not indicated if the data were tested for normality (and which test was used).
Normal distribution of data was investigated using the Kolmogorov-Smirnov test. We added this information in the Materials and Methods section (line 282).
“Normal distribution of data was investigated using the Kolmogorov-Smirnov test.”
I do not believe n=3 should be tested with a parametric test.
Please see our answer above. We usually analyze data from RT-PCR analyses with 3-4 animals per group using a parametric test. Statistical analysis of data with sample sizes of 3-4 per group using a parametric test is also practiced by other research groups (e.g., Yim AKJ et al., Nat Neurosci 2022; Peck LJ et al., J Neurosci 2021; Yu X et al., Nat Commun 2020).
To clearly state the total number of animals used.
The number of animals used for every experiment is indicated in the figure legends. Moreover, we now provide the total number of animals used for this study in the Materials and Methods section, line 79: “In total, 178 mice were used in this study. ”
It doesn’t seem that the authors took into account the issue of the splice variant of mouse TRPA1 (https://doi.org/10.1038/ncomms3399). Please revise and discuss this point.
Thank you very much for this suggestion. We now discuss the issue of the splice variant of mouse TRPA1. Line 547: “It should be also considered that a TRPA1 splice variant has been identified that may interact with full-length TRPA1 and alter its expression at the plasma membrane, thereby theoretically affecting the results of tissue staining experiments [63].”
A TRPA1 antagonist should have been used to confirm the obtained data.
We have performed additional patch-clamp experiments with a TRPA1 antagonist. Please see our answer to reviewer #1.
In the discussion section poor attention was given to the behavioral data. For instance, the data reported in Figure 1B and Figure 2G show a different pattern (in terms of time) of behavioral response after mechanical stimulation in Slack-/- and SNS-Slack-/-.
We now discuss the behavioral data in the discussion section, line 570: “It is also worth noting that the alteration in AITC-induced mechanical hypersensitivity was more pronounced in Slack-/- mice (significantly different from WT mice 3 h to 48 h after the AITC injection) than in SNS-Slack-/- mice (significantly different from control mice only at 24 h after the AITC injection). It seems therefore possible that Slack channels expressed in NaV1.8-negative sensory neurons (in which Slack is not knocked out in SNS-Slack-/- mice) or in the central nervous system might also contribute to TRPA1-mediated pain processing.”

Round 2
Reviewer 1 Report
I thank the authors for all the corrections they have carefully made to the manuscript. In my opinion, the manuscript has been sufficiently improved and can be accepted in its present form.
Reviewer 2 Report
The revised version is fine.